# Xylem Embolism and Pathogens: Can the Vessel Anatomy of Woody Plants Contribute to *X. fastidiosa* Resistance?

**DOI:** 10.3390/pathogens12060825

**Published:** 2023-06-12

**Authors:** Giambattista Carluccio, Davide Greco, Erika Sabella, Marzia Vergine, Luigi De Bellis, Andrea Luvisi

**Affiliations:** Department of Biological and Environmental Sciences and Technologies, University of Salento, 73100 Lecce, Italy; davide.greco@unisalento.it (D.G.); erika.sabella@unisalento.it (E.S.); marzia.vergine@unisalento.it (M.V.); luigi.debellis@unisalento.it (L.D.B.); andrea.luvisi@unisalento.it (A.L.)

**Keywords:** cavitation, biotic and abiotic stresses, cell wall, pit

## Abstract

The maintenance of an intact water column in the xylem lumen several meters above the ground is essential for woody plant viability. In fact, abiotic and biotic factors can lead to the formation of emboli in the xylem, interrupting sap flow and causing consequences on the health status of the plant. Anyway, the tendency of plants to develop emboli depends on the intrinsic features of the xylem, while the cyto-histological structure of the xylem plays a role in resistance to vascular pathogens, as in the case of the pathogenic bacterium *Xylella fastidiosa*. Analysis of the scientific literature suggests that on grapevine and olive, some xylem features can determine plant tolerance to vascular pathogens. However, the same trend was not reported in citrus, indicating that *X. fastidiosa* interactions with host plants differ by species. Unfortunately, studies in this area are still limited, with few explaining inter-*cultivar* insights. Thus, in a global context seriously threatened by *X. fastidiosa*, a deeper understanding of the relationship between the physical and mechanical characteristics of the xylem and resistance to stresses can be useful for selecting *cultivars* that may be more resistant to environmental changes, such as drought and vascular pathogens, as a way to preserve agricultural productions and ecosystems.

## 1. Introduction

In tracheophytes, the xylem is in charge of transporting sap from roots to leaves. The sap forms in the xylem lumen a liquid column, thanks to its cohesive properties [1], and it is subjected to a negative pressure in the range of −0.4 and −8 MPa [2] due to transpiration [3]. This strong negative pressure makes the sap metastable [2,3,4], and so it tends to suddenly vaporize to achieve stability [1]. The stability can be reached after an external stimulus that triggers a process known as cavitation; it leads to the formation of air bubbles called emboli, which break the water column, causing blockage of the sap flow in the vessel [5]. The formation of emboli causes a loss of the conductivity and hydraulic functionality of the xylem, and in severe cases, it can compromise the survival of the plant. Despite the negative impact of cavitation on the plant, the xylem’s overall conductance varies, depending on multiple factors, so that plants live with a variable portion of the xylem not entirely functional because of cavitation [6]. However, plants have developed structural systems capable of limiting the cavitation phenomena and the effects associated with it [7], allowing the colonization of a wide range of habitats, especially xeric ones [8]. For example, the stomatal closure before the cavitation point [9], the compartmentalization of the xylem, the presence of bordered pits, and the renewal of the xylem tissue in each vegetative season are considered protection strategies [8,10]. Parallel to these adaptations, refilling mechanisms are also essential to restore the functionality of the vessels once cavitation has occurred [11]. In general, the factors that can commonly induce cavitation are water deficit, thermal stress, and attacks by pathogens [12,13]; currently, plants are subjected to new challenges due to global warming, and humans’ activities expose plants to extreme abiotic stresses and new pathogens. For instance, *Xylella fastidiosa* [14] is an emerging pathogen in Europe that can cause many diseases in a wide range of crops and wild plants, such as phony peach disease, plum leaf scald, Pierce’s disease, citrus variegated chlorosis, and other types of scorch syndromes on almond, coffee, and oleander [15,16,17]. Thus, it poses a serious threat to ecosystems and agriculture since perennial woody plants present in a natural or anthropized context can provide numerous ecosystems and economic and landscape services [18,19], counteract climate change [20,21], reduce soil erosion, and provide a wildlife habitat [22,23]. As reported later, studies indicate that xylem embolism can have a role in *X. fastidiosa* pathogenicity. Therefore, understanding which phenomena can cause cavitation, which are the predisposing factors of how emboli develop in vessels and how plants cope with these issues, is useful for the implementation of forestry and agricultural management strategies, particularly in a context in which globalization of harmful vascular pathogens and the effects of climate change stress undermines the stability of ecosystems and agriculture.

### 1.1. Role of Abiotic Stress in Cavitation

Drought is the most common cause of cavitation because the rise in water deficit causes an increase in negative pressure in the xylem vessels [24]. The strong negative pressure can suck into the lumen of the vessel the air bubbles that may be present in the porous areas of the cell wall, so the bubbles in turn trigger the phase change of the water present inside the vessel, increasing the size of the embolus. This mechanism is known as “air seeding” [25,26]; the excessive negative pressure during intense water stress can lead to a loss of functionality of the pit membranes, while the air bubbles can also propagate in the adjacent vessels [27]. However, plants differ in susceptibility to cavitation, and it has been shown that high resistance to this phenomenon is linked to higher tolerance to water deficit, as reported by Pockman et al. [28] for riparian and upland Sonoran Desert species. These authors established that there is a certain predictivity in the tendency towards cavitation; for example, the average pressure value of a xeric plant xylem would cause cavitation in plants native to more humid environments [28]. Another cause of cavitation can be freezing, as the ice formed can block the sap flow in the vessel [29]; this can happen especially during early spring when the sap begins to thaw but some is still frozen [30]. In addition, the formation of air bubbles during water freezing is favored by the fact that air is insoluble in ice, and as demonstrated by observations carried out on angiosperms and gymnosperms [31,32], at the time of thawing, bubbles trapped in the ice can cause emboli [33,34]; further, it is thought that larger and longer vessels may contain larger bubbles in higher numbers, so such vessels have a greater probability of incurring embolisms than narrower and shorter vessels [34,35]. In fact, it seems that xylem vessels with a diameter of less than 30 μm are less predisposed to embolisms caused by freezing [36]. Nevertheless, studies conducted on fir trees (*Picea abies* L.) have shown that embolism can affect even the smallest vessels if repeated cycles of freezing–thawing occur [37].

In a context of climate change, extreme temperature episodes ranging from hot to cold waves are expected [38,39,40], so with a wide range of temperatures and erratic rain patterns, an increased incidence of cavitation episodes, damage to vascular structures, and altered functions, all resulting in hydraulic failure and carbon starvation [41,42], can be assumed. Drought and freezing episodes can also impact plants’ physiology and ecology, disease incidence, and insects’ behavior, causing a general plant decline [43,44,45,46,47,48,49]. This trend can lead to a decrease in the suitability of lands to crop productivity, representing a threat to food chain and security [50,51], as extreme temperatures and drought are the main factors of yield losses [52].

### 1.2. Role of Biotic Stress in Cavitation

The presence of pathogens in the xylem can cause embolism and reduced vessel conductivity, often inducing wilting of plants [53,54]. The ability of a pathogenic organism to interfere with the proper functioning of the xylem is related to its ability to invade the host’s tissues, and it may be due to several factors that intervene in different ways: by lowering the surface tension of the sap, as with the pathogen *Endoconidiophora polonica* (Siemaszko) [55,56]; by modifying or destroying the structure of pit membranes [57], as observed with *X. fastidiosa* during transverse movement between vessels [58,59]; by perforation of vessel walls [60]; or, as in the case of the nematode *Bursaphelenchus xylophilus* (Nickle), by producing volatile and hydrophobic compounds [61]. In some wilt diseases of oak [13], pine [62], grapevine [63], and olive trees [64], the association between pathogen, vessel embolism, and development of symptoms is strongly supported, as well as the role of air bubbles to trigger the defensive response of the plant, resulting in vessels’ occlusion by the production of a gel and tylosis. Furthermore, embolism is considered the main cause of loss of xylem conductivity, while occlusion is a subordinate event [65]. Nevertheless, the association between embolism and wilt disease may be not always be correlated, as reported for esca disease of grapevine [66].

## 2. Wood Anatomy, Cavitation, and Resistance to Pathogens

Resistance to cavitation is a complex characteristic determined by numerous interconnected factors not always consistent among the different botanical species in which this resistance has been observed (Figure 1).

Cavitation can vary among tree species; for example, deciduous and evergreen plants show a similar attitude to it, while tropical plants are more susceptible than species typical of temperate climates [67]. These differences are also present at the intraspecific level [68,69], depending on the genotype [70,71], and among the organs of the same plant [72,73]. For example, it has been observed on beech (*Fagus sylvatica* L.) that branches exposed to direct sunlight are more resistant to cavitation than shaded ones [74], so in the face of this variability, it is not easy to affirm with certainty and fullness which are the traits that make plants less or more prone to cavitation. However, in woody plants, the tendency towards cavitation phenomena appears to be influenced by the interaction among xylem tissue, vessel network and pit characteristics, vessel organization [75], number and diameter of pits in a vessel [25,76,77], ultrastructure and structure of pits and its components [75], and characteristics of a vessel, such as wall thickness and length [75,76,77]. For instance, in gymnosperms, it has been observed that resistance to embolism is due to the flexibility of the pit membrane and the large size of the torus compared with the pit aperture, which thus guarantees adequate adhesion of the torus and an efficient sealing of the vessel [78], but because the characteristics of vessels and pits may vary in different organs of a plant, the vulnerability of plant organs to cavitation may be different [79]. Moreover, gymnosperms generally have a lower quantity of parenchyma cells than angiosperms, in which parenchyma cells can constitute 80% of the xylem tissue; for this reason (and due to the larger and less dense vessels), angiosperms are considered more vulnerable to cavitation, although parenchyma cells could play a fundamental role in cavitation recovery since nonstructural sugars used for refilling are stored here [80,81]. Furthermore, ring-porous trees—in which the vessels produced at the beginning of the growing season are larger than the vessels formed at the end of the season—are more susceptible to freezing cavitation than trees that have small and homogeneous xylem vessels (diffuse-porous trees) [29]. Despite this, vessels of different sizes are advantageous during embolism recovery as small vessels would offer a reservoir of water for refilling adjacent larger vessels that are more prone to cavitation [35]. Wider xylem vessels are often associated with increased susceptibility to embolism [82] as they are not only rich in pits [83], but air bubbles also tend to expand more easily [18], while narrow vessels not only increase the plant’s resistance to drought but also make the plant less prone to cavitation [84].

Since cavitation is a common event, it is important for the plant to activate mechanisms that allow the recovery of vessels functionality and sap flow, that is, “refilling” [85,86]. The refilling process is not completely understood, but it was reported that the depletion of sugars accumulated in the xylem parenchyma may play a role in these responses [85,87,88]. Again, the ability of the plant to recover the full functionality of vessels seems to be correlated with the organization of the xylem; this may explain why some species with low wood density can be more efficient in hydraulic recovery than species with higher wood density [81,89], showing a linear correlation between the amount of parenchyma and the amount of nonstructural sugars accumulated in the stem [90,91,92]. Although the phenomenon of refilling has not been observed in all plants [93], it is thought that the starch-accumulating parenchyma cells can release sugars to the vessels, generating an osmotic force, which draws water into the lumen of the vessel [94,95]. Anyway, in the literature, there are also indications supporting the hypothesis that, in addition to sugars, other molecules, such as ions and proteins, are involved in the refilling process [96,97], with pit membranes regulating the movement of solutes in the lumen of cavitated vessels [98]. Another aspect involves the physical and chemical characteristics of the xylem tissue that might predispose plants to tolerance or susceptibility to vascular pathogens; sure enough, these different characteristics can affect the process of infection and colonization by pathogens, facilitating or containing it so that, in some cases, the xylem constitutes the first plant defense against infection [99] (Figure 2).

It has been shown that some grape *cultivars* with a higher number of vessels with a diameter larger than 120 μm are more susceptible to infection of *Phaeomoniella chlamydospora* (Crous and Gams) [103,104] (Figure 2A). The same trend has been reported for elm in which the spread of *Ophiostoma ulmi* (Buisman) and the severity of infection symptoms are due to the configuration of vessels. In addition, it was observed that resistant species and varieties have a higher density of shorter and narrower vessels and smaller bordered pits and a lower number than susceptible ones [105,106,107] (Figure 2B). Even on avocado trees (*Persea americana* Mill.), it was found that the *cultivar* susceptible to the fungus *Raffaelea lauricola* (Harr., Fraedrich and Aghayeva) possesses vessels with a larger average diameter than the resistant *cultivars*, but no significant differences were found in vessel aggregation and density [102] (Figure 2C). The hypothesis is that smaller vessels are quicker compartmentalized by the host with gums and tyloses, and a small number of pits with a lower area can limit the spread between vessels of the pathogen [108]. Another factor that can influence the diffusion of pathogens and of emboli is the frequency of vessel terminations, especially at the leaf–stem junction, where some vessels close to flow into others, continuing towards the leaf [25,109]. This conformation represents a sort of protection system as most of the bubbles and particles are filtered by a pit membrane, even if the few open vessels that run from the leaf to the stem can play a huge role in the passive spread of pathogens and gas bubbles [110,111,112,113] (Figure 2D).

## 3. Embolism and Xylem Anatomy May Be Involved in Vulnerability to *X. fastidiosa*

*X. fastidiosa* is a plant pathogenic bacterium that belongs to the *Xanthomonadaceae* family and shows a wide range of host plants [114], including more than 650 species [115]. It is a systemic bacterium that colonizes exclusively the xylem tissue, where it finds nourishment and where it replicates [116], causing scorching and stunting diseases worldwide [117] in numerous crops, such as olive trees (*Olea europaea* L.), citrus trees (*Citrus* spp.), grapevine (*Vitis* spp.), and stone fruit (*Prunus* spp.), and forest trees, such as oaks (*Quercus* spp.), maple (*Acer* spp.), sycamore (*Platanus* spp.), and elm (*Ulmus* spp.) [117,118].

As for the pathogenesis, it is commonly accepted that the main factor causing the symptoms is the blockage of the xylem vessels due to the bacteria biofilm and to the active plant production of tylosis and gums [119], which cause hydraulic dysfunctions [119,120]. However, the development of symptoms may not be entirely due to the vessel’s occlusion, so the pathogenesis may be more complex, and it may take place from the earliest stages of the infection, before the actual colonization of vessels occurs [13] as sustained in other pathosystems [56,65] (Figure 3).

With this conclusion came two studies carried out on oaks and grapevine, in which it was shown that embolism occurs before colonization, during the early stage of infection by *X. fastidiosa* and preceding the appearance of symptoms [13,63]. Still, McElrone et al. [13] described that for *Quercus palustris* (Münchh) and *Q. rubra* (L.), the conductivity of the xylem decreases progressively with the spread of *X. fastidiosa* until it disappears completely, after which the symptoms of infection occur. Particularly, Cochard and Tyree [121] highlighted that in oak, embolism is a necessary condition for triggering a plant’s defense response that consists in occlusion of the vessel with tylosis to isolate the bacterium in the infected vessel [122]. Moreover, *X. fastidiosa* can produce cell-wall-degrading enzymes [123], and it has been hypothesized that emboli formation may be caused during the lateral movement of *X. fastidiosa* through the destruction of the bordered pits with a consequent negative effect on the xylem conductivity [58,124], so the presence of air bubbles and sugars (needed for vessels’ refilling after embolism) would promote the proliferation of *X. fastidiosa* due to the nutrient-rich and aerobic conditions [124,125]. Moreover, it was suggested that the host response to the pathogen is related to the xylem vessel traits and to the pit membrane’s structural qualities, such as size, resistance to enzymatic degradation, and frequency [124] reporting, for example, that the wide pits found in oaks would facilitate the passive migration of *X. fastidiosa* cells, which measure 0.25–0.35 µm in diameter and up to 4.0 µm [14] in length, making oaks more vulnerable to cavitation [13].

In some areas of the world, *X. fastidiosa* is an emerging plant pathogen that causes diseases and tree deaths, producing huge cultural and economic losses [126]. Since there are no therapies for infected plants, one possibility for disease management may be understanding how the morphological traits of the xylem can influence the spread of the pathogen and the plant’s response to infection. This might suggest selecting woody plants for the anatomical xylem features of interest with the aim of increasing the plant constitutive defenses. For this purpose, the literature reporting the relationship between functional xylem anatomy and response to *X. fastidiosa* in some woody plants will be reviewed.

### 3.1. Grapevine

*Vitis vinifera* L. is a plant species that counts about 20,000 *cultivars* [127], presenting great variability in xylem features owing to different domestication environments: *cultivars* from semiarid areas have a wider vessel diameter than *cultivars* from temperate areas [128]; the wood can be ring porous or diffuse porous; and the average vessel is 3 or 4 cm long [129], even if vessels up to 1 m have been observed [12], while vessels can have a diameter up to 200 µm [130]. Xylem attributes seem to be indispensable for grapevine defense against emboli induced by drought [131], as confirmed by Brodersen et al. [132], who evaluated two *cultivars*, the drought-resistant Shiraz and the drought-sensitive Grenache, observing a different distribution of xylem vessels, with the Grenache *cultivar* showing a higher tendency towards vessels embolism because of the greatest number of large vessels (diameter between 120 and 150 µm) in comparison with Shiraz.

Studies on grapevine vessel diameter have also been carried out in relation to Pierce’s disease [133,134,135], evaluating grapevine *cultivars* showing different responses to infection by *X. fastidiosa* (Table 1) [110]. Significant differences were not reported in relation to petioles, where vessel diameter was similar regardless of *cultivar*, with 70% ranging between 10 and 45 μm; instead, in the stem, the tolerant *cultivar* Sylvaner had vessels with a smaller diameter ranging mostly between 85 and 100 μm, while susceptible *cultivars* displayed vessels between 150 and 300 μm. Furthermore, the fact that the Sylvaner *cultivar* had 20% more parenchyma rays than the other *cultivars* led to the assumption that this tissue organization may play an additional role in contrasting the spread of pathogens and emboli by secreting antimicrobial compounds able to limit the transverse movement of *X. fastidiosa*, which represent a barrier to pathogen movement [136].

In addition, host plants that allow greater movement of the pathogen have more connections between the stem and leaves than hosts in which the movement of *X. fastidiosa* is limited, as indicated by Deyett et al. [139], who compared grapevine *cultivars* susceptible to *X. fastidiosa* with the resistant wild species *V. arizonica* (Engelm). They showed that vascular characteristics influence susceptibility to infection, as susceptible *cultivars* had a significant decrease in xylem conductivity upon infection along with a wider and faster circulation of the bacteria than the resistant grapevine species. In grapevine, the spread of bacteria depends also on the age of tissues, because an older structure, with an abundant secondary xylem mainly characterized by bigger vessels and scalariform pits, weakly counteracts the spread of bacteria in comparison with the primary xylem of younger structures, which is characterized by smaller vessels [140]. Those differences influence pathogenesis, causing a larger number of infected vessels and a higher extent of colonization of these vessels in comparison with more resistant *cultivars* [120,141]. Other works showed that numerous wild American *Vitis* plants are resistant to Pierce’s disease in contrast with commercial *cultivars* [142,143], and that *X. fastidiosa* spreads widely in the xylem of susceptible grapevine plants while remaining near the inoculation point in resistant plants [144,145]. This may depend on the ability of the pathogen to hydrolyze pit membranes as it is influenced by their chemical composition and, in particular, by the proportion of weakly and heavily methyl-esterified homogalacturonans and fucosylated xyloglucans. These factors can explain the different ability of the bacteria to spread in the xylem tissue and the plant’s varied response to infection. Thus, the limited movement of bacteria cells in resistant grapevines, unlike susceptible ones, can be due to the maintenance of the integrity of pit membranes. These findings suggest that finer differences, such as cell wall thickness, the level of pitting, and features of pits, may play a key role in resistance. However, to define paths of resistance, it is necessary to extend observations to more vine varieties, also considering that the response to infection could also belong to other inducible or noninducible factors that result in resistance [146].

The observations made on grapevines about the relationship between the cytological and histological characteristics of the xylem and the response of the plant to the *X. fastidiosa* infection can also be supported through comparisons with other pathogens. Those results strongly support the existence of a relation between xylem features, such as density and diameter, and the ability of pathogens to colonize xylem tissue. This confirmed the data obtained by Pouzoulet et al. [103], who have observed that the *P. chlamydospora*–susceptible *cultivar* Thompson Seedless had a wider vessel diameter (>120 μm) than the resistant *cultivar* Merlot, which presents a mostly narrower vessel (from 80 to 119 µm).

### 3.2. Olive Tree

It is well known that olive tree xylem sap can withstand high tensions when severe drought occurs without embolization [57,147] even if different *cultivars* can display different vulnerabilities to drought stress [137,148]. De Micco et al. [149] reported that olive trees have xylem vessels evenly distributed and homogeneous in the ring of the year. The mean diameter of stem wood vessels is approximately 45 μm [149] generally, no longer than 8 cm, even if longer ones may be present [150]. Vessels are distributed in radial and diagonal rows, and they have a simple perforation plate, small and alternate [151].

In olive trees, hydraulic architecture plays an important role during water stress, and some *cultivars* can manage drought by reducing vessel diameter and increasing vessel density during drought episodes [152], but the effects of different water regimes on xylem vessels may not be equal or significant in all olive *cultivars* [153]. The relation between vessel anatomy, drought resistance, and cavitation in olive trees was shown by Trifilò et al. [154], studying the Leccino Dwarf (LD) and Leccino Minerva (LM) *cultivars* and their grafting combinations. LM with its larger vessels and higher vessel density was more prone to cavitation induced by drought than LD, and when LD was grafted on LM, the rootstock was able to modify the xylem architecture in the scion, which developed wider vessels than an ungrafted LD plant. For instance, vessels with a diameter above 25 µm were 26% in the grafted LD scion and only 12% in the ungrafted LD, while vessels with a 30–40 µm diameter were 12% in the grafted LD and absent in the ungrafted LD. Such modifications decreased the plant’s resistance to water stress, additionally increasing the susceptibility to cavitation of the LD scion. Similar results were obtained by Ennajeh et al. [148] with the olive *cultivars* Chemlali and Meski, in which the resistance to cavitation in olive trees was positively correlated with drought resistance.

The *X. fastidiosa* subspecies *pauca* strain “De Donno” is responsible for the Olive Quick Decline Syndrome (OQDS) [16], in which the *cultivars* Cellina di Nardò and Leccino show different responses to infection [155], with Cellina di Nardò being considered highly susceptible to infection and Leccino being reported as resistant [156]. Such variability in response could be due to cyto-histological differences in the xylem [124] because although the maximum length of the xylem vessels is similar in both *cultivars* (about 80 cm) [124], Leccino has constitutively a greater number of xylem vessels, which are more homogeneous and have a smaller diameter (maximum of approximately 30 μm). Conversely, Cellina di Nardò is characterized by vessels that can measure up to 75 μm in diameter [157], which are present in smaller numbers in respect to the total supplied leaf mass compared with Leccino (Table 1) [124]. Moreover, the density of vessels is constitutively higher in Leccino (40.7 ± 2.2 vessels per mm^2^) than in Cellina di Nardò (36.6 ± 2.8 vessels per mm^2^) [157], suggesting that the higher susceptibility to *X. fastidiosa* shown by the *cultivar* Cellina could be due to the larger diameter of the xylem vessels that can facilitate the spread of the bacteria [123] by predisposing the plant to higher embolism vulnerability [157]. Thus, although large xylem vessels are more efficient in terms of hydraulic conductance, making Cellina di Nardò drought resistant compared with Leccino, they are more prone to cavitation phenomena [137,158]. It should also be noted that in Cellina di Nardò, water stress causes limited effects (e.g., on genes responsive to plant water status and pathogen infection, or on sugar content) when combined with *X. fastidiosa* infectious status [137].

In relation to *X. fastidiosa* resistance, another interesting potential olive *cultivar* is SX32, which was genetically correlated with the *cultivar* Chemlali [159]; a comparative study among 11 olive *cultivars* revealed that the *cultivar* Chemlali has a xylem tissue with a high area and high density and with one of the smallest vessel diameters among the observed *cultivars*, between 10 and 15 µm [160]. Even if, for the olive tree, there is a strong indication of the relationship between small vessel diameter and resistance to cavitation/resistance to vascular pathogens, other factors that could contribute to resistance to *X. fastidiosa* [161] are the characteristics of the pit membranes; indeed, it was found that *X. fastidiosa* can destroy the pit membranes more extensively in Cellina than in Leccino, probably due to the different polysaccharide composition (as homogalacturonans, xyloglucans, and cellulose) of the membranes [121,146]. To understand the resistance mechanisms of the olive tree or the causes of susceptibility, how important it is to evaluate the differences in the composition of the xylem in its various chemical–physical aspects in several *cultivars* emerges in order to be able to verify the presence of trends or common aspects between the *cultivars* of interest.

### 3.3. Citrus

The *X. fastidiosa* subsp. *pauca* causes Citrus Variegated Chlorosis (CVC) [162,163] in plants of the *Citrus* genus. One of the most susceptible citrus trees species, *Citrus sinensis* (L.), usually presents solitary or grouped vessels with a diameter between 60 and 90 µm, simple perforation plates, and small pits [130]. Regarding the relationship between the physical characteristics of the xylem tissue and the tendency of the vessels to cavitate, there is no information in the scientific literature; however, some data have been found on the relationship between xylem and response to the attack of vascular pathogens.

Coletta-Filho et al. [138] evaluated the stem xylem anatomy of several citrus hybrids (*C. sinensis* × *C. reticulata*) with different responses (susceptible, tolerant, and resistant) to *X. fastidiosa*; they reported that, despite the differences observed, there was no correlation between the diameter or density of vessels and response to bacterial infection (Table 1). Indeed, it was found that susceptible plant hybrids have narrower xylem vessels compared with tolerant genotypes, but the susceptible *cultivar C. sinensis* has larger vessels than its resistant hybrids. An identical conclusion was obtained by Garcia et al. [164] after the analysis of 11 *Citrus cultivars*’ response to *X. fastidiosa* infection; they did not find a clear difference in the leaf petiole xylem in terms of number of vessels and structure despite the diversity in the symptoms. Thereafter, Niza et al. [165] studied the *X. fastidiosa* colonization in the stem and petioles of susceptible and resistant genotypes of citrus, showing that the bacteria spread widely in the petioles of both susceptible and resistant genotypes. Furthermore, in the stem, they found that the bacteria were able to colonize the primary and the secondary xylem of susceptible *cultivars*, while in the resistant *cultivar,* only the primary xylem was colonized despite no significant differences in cell wall deposition and vessel distribution; this suggests that the resistance in plants of the genus *Citrus* is not related to the anatomical traits of vessels [164], but other mechanisms could be involved, such as secretion of chemical compounds (as phenolic compounds) and, in particular, lignin deposition that can restrict the spread of bacterial cells in primary xylem vessels [146,165,166]. Resistant plants exhibited a quicker and more pronounced reinforcement through lignification of vessels, explaining the different tendency to spread in the xylem tissue of *X. fastidiosa* in susceptible and resistant citrus plants [165].

## 4. Methods for Measuring Vulnerability to Cavitation

The variety of methods available for measuring cavitation in the xylem reflects the necessity of an accurate and reliable technique to study different plant materials. Each technique presents advantages and disadvantages given the fact that the xylem sap, being in a metastable state, is highly sensitive to perturbations, which can cause both the entry of air into the xylem ducts and the dissolution of air bubbles already present by spoiling the experimental observation [167]. However, the various techniques can be classified into three categories according to the principle on which they are based.

### 4.1. Acoustic Techniques

These are qualitative methods based on the observation that during cavitation, part of the energy is dissipated as sound frequency that can be detected with different types of sensors, depending by the range of frequencies [168,169]. The advantage of these techniques is that they are not invasive on plants [167]; however, the acoustic emissions are not exclusive to cavitation events [170], so other accidental events producing sound can affect the results.

### 4.2. Visual Techniques

These are qualitative methods based on the direct observation of the emboli in the vessels, and they can be carried out with different technologies, such as electron microscope, X-rays, or magnetic resonance [171,172,173]. Visual techniques, such as magnetic resonance imaging, are nondestructive tools for assessing in vivo *X. fastidiosa* behavior and the effects on water transport [174]. The main drawback is related to the sample preparation since air bubbles can quickly dissolve or create new ones. Furthermore, the necessary equipment is often not suitable for routine observations, and it requires skilled researchers [167]. Alternatively, indirect observation of the cavitated vessels with dyes, such as safranin or basic fuchsin, is easier to operate. However, direct and indirect observations are complementary as indirect methods are often preliminary to detect plant tissue sections with emboli [167].

### 4.3. Hydraulic Detection of Embolism

This is a commonly used and quantitative technique based on the evaluation of xylem hydraulic conductivity through the measurement of the conductivity of the xylem with emboli when it is completely saturated with water [167,175]. The disadvantage is that the quantity of emboli can change in the time between sampling and measurement [167].

## 5. Conclusions

The length, diameter, and other characteristics of the xylem vessels are influenced by plant genetics and environmental factors [176], such as water availability and temperature [177,178]. If large xylem vessels, as shown by Poiseuille’s law, guarantee greater hydraulic conductivity and increase the efficiency of water transport [176], on the other side, they are more predisposed to embolism and also more vulnerable to vascular pathogens [25,108]; vessel diameter can be substantial in the interaction between vascular pathogens, such as *X. fastidiosa*, and the trees. Nevertheless, the differences observed in the response to *X. fastidiosa* (or to other pathogens) in several host plants depend even on the intrinsic variance of the hosts [124]. This confirms the importance of enhancing knowledge of the relationship between xylem structure and susceptibility to pathogens to find valid solutions to [179] contrast diseases that affect plants of agricultural and forestry interest [179]. Even if in some pathosystems the relationship between xylem structure and susceptibility to vascular pathogens is well documented (such as in elm [107,180] and grapevine [103,125]), for other host plants, this correlation is weakly sustained due to several reasons: the lack of literature on emerging pathogens, the remarkable inter- and intraspecific diversity of plants, the environmental conditions that can modify the structure of xylem vessels [181,182], and the pathogens’ variability. However, it is known that plants evolve their defenses against vascular pathogens [183], such as the ability to confine microorganisms and their eventual toxic compounds via compartmentalization [102,125]. Among the many features of a xylem vessel, diameter is the main element that can influence tolerance or susceptibility to wilt diseases [125] also because a low amount of tyloses [104] (and energy) is required for pathogen compartmentalization in small vessels [125].

In areas where *X. fastidiosa* is an emerging pathogen [184,185,186], a wide spectrum of responses among plants can be observed, indicating that the pathogenesis of *X. fastidiosa* depends on many variables, which are both intrinsic and extrinsic to the infected plants [115,124,187], and as reported here, the anatomical characteristics of xylem vessels, such as diameter, number of vessels, and type of pits, may play an important role in the defensive response, at least for grapevine and olive; in addition, this tendency was not observed in citrus trees.

Due to their economic value, plants such as olive trees, grapevines, and citrus trees are studied more than other woody species, but they represent only a small part of possible hosts and, consequently, few of a great variety of interactions that *X. fastidiosa* can determine. Thus, the lack of multispecies (and multi-*cultivar*) observations makes it difficult to identify common resistance traits useful for developing a control strategy based on reliable patterns concerning pathogenesis, such as the correlation between anatomical features of vessels, their organization in the xylem tissue, and susceptibility or tolerance. In addition, research should be focused more on the composition of the cell wall and pits and on the mechanisms underlying cavitation. In order to study the mechanisms underlying cavitation and the role of abiotic and biotic stresses, it is essential to develop new methods that give reliable and repeatable results, because the actual techniques are not considered to be devoid of significant bias [167].

Finally, an in-depth knowledge of the anatomy of woody plants in relation to embolic phenomena and *X. fastidiosa* could find application in the context of breeding programs, which have already led to satisfactory results for other vascular pathogens, such as *Fusarium xylarioides* and *Ceratocystis fimbriata* [188,189]. Phenotyping could strengthen the link between xylem features and pathogen resistance, identifying the biological markers for resistance to vascular pathogens and driving the research activity towards the selection of species and *cultivars* resistant to *X. fastidiosa* and how it is happening in breeding olive programs for resistance to *Verticillium dahliae* [190,191,192]. Traditional breeding could be also supported by other biotechnological methods, such as priming, showing positive results in terms of symptom development and resistance-related gene expression on grapevine infected with *X. fastidiosa*, where grapevines primed with lipopolysaccharide developed fewer tyloses compared with nonprimed plants, reducing occluded xylem vessels and symptoms [193].

## Figures and Tables

**Figure 1 pathogens-12-00825-f001:**
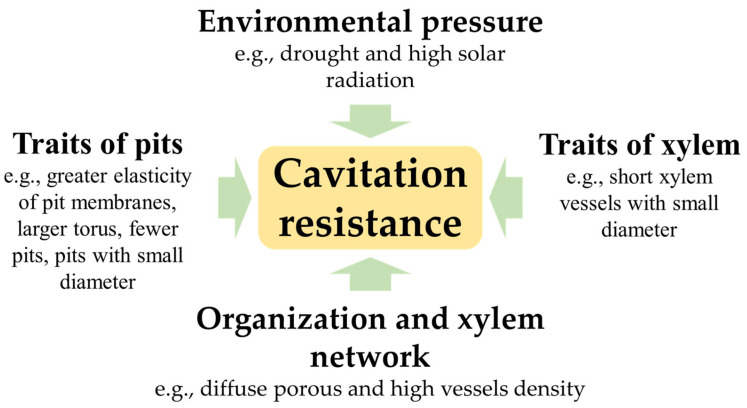
Factors that determine resistance to cavitation, which are many and often differ according to the plant species and the context in which they are observed. However, it can be assumed that plants characterized by short xylem vessels, with a small diameter and with small and few pits, are more resistant to cavitation.

**Figure 2 pathogens-12-00825-f002:**
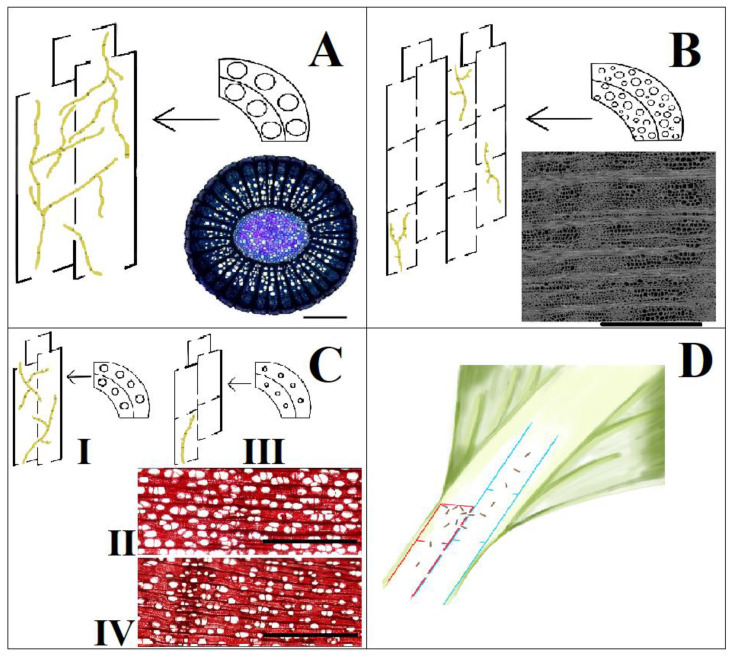
Schematization of the pathogen spread (represented in brown color) in the xylem tissue of susceptible or resistant plants: (**A**) xylem of the *Phaeomoniella chlamydospora*–susceptible grapevine *cultivar* Thompson Seedless, characterized by wide and long xylem vessels (photo from [100], bar = 1 mm); (**B**) xylem of the *Ophiostoma ulmi*–tolerant elm hybrid Dodoens, capable of a fast reaction to infection that includes the formation of much smaller vessels and an increase of their density (photo from [101], bar = 500 μm); (**C**) comparison of the xylem between the *Raffaelea lauricola*–susceptible avocado *cultivar* Simmonds (I and II) and the resistant one Duke (III and IV), in which besides differences in vessel length and diameter (C I and C III), any significant difference in the vessel’s density and organization is reported (photo from [102], bar = 1500 μm); (**D**) vessel ending (red) at the petiole level, which acts like filter for emboli and pathogens, especially at the junction between leaf and stem; adjacent long and open vessels (blue) can facilitate the circulation of emboli or pathogens.

**Figure 3 pathogens-12-00825-f003:**
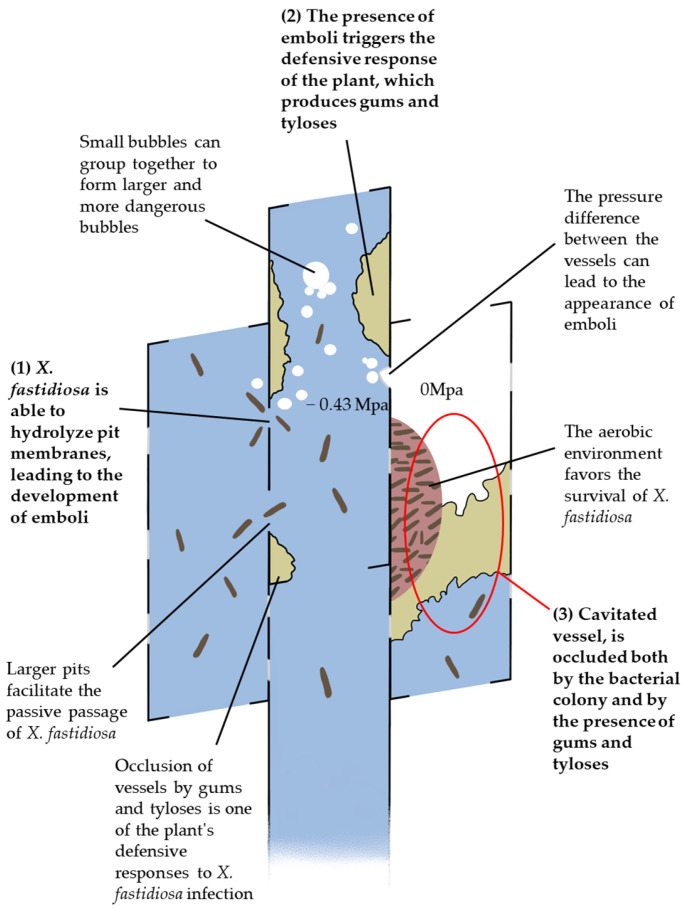
The cavitation process in xylem vessels infected by *X. fastidiosa*: (**1**) *X. fastidiosa* can move actively between vessels by hydrolyzing pit membranes; (**2**) this can lead to the production of emboli, which seem to stimulate the defensive response of the plant that starts to produce gums and tyloses, as well as those induced by the pathogen; (**3**) in the most advanced infection stages, gums, tyloses, and bacterial colonies can occlude the vessels, causing cavitation, which favors the survival of the pathogen.

**Table 1 pathogens-12-00825-t001:** Xylem anatomy of some cultivated woody hosts of *Xylella fastidiosa*.

Plant and *Cultivar*	Attitude Towards *X. fastidiosa*	Vessel Diameter (µm)	Other Factors
Grapevine ^(1)^			
*Sylvaner*	Tolerant	80–250	20% more of parenchyma rays ^(1)^
*Cabernet* *Sauvignon*	Susceptible	150–400	
*Pinot Noir*	Highly susceptible	150–400	
*Chardonnay*	Highly susceptible	150–400	
Olive tree			
*Leccino*	Resistant	<30 ^(2)^	Presence of starch grain in vessels ^(2)^
			Compact pits ^(3)^
			Occlusion with callose like structures ^(3)^
*Cellina di Nardò*	Susceptible	45–75 ^(3)^	
Citrus tree ^(4)^			
*Hybrid H124*	Resistant	20	
*Hybrid H 34*	Resistant	21	
*Tangor Murcott*	Resistant	27	
*Hybrid H155*	Susceptible	15	
*Hybrid H 179*	Susceptible	19	
*Pera sweet orange*	Susceptible	23	

^1^ [110] ^2^ [137] ^3^ [121] ^4^ [138].

## Data Availability

Not applicable.

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
