# Peer review of "Xylem Embolism and Pathogens: Can the Vessel Anatomy of Woody Plants Contribute to X. fastidiosa Resistance?"

_pathogens, 2023, doi:10.3390/pathogens12060825_

Round 1
Reviewer 1 Report
The manuscript from Carluccio and colleagues is a review about the role of xylem embolism in the pathogenesis of xylem-limited pathogens with particular emphasis to Xylella-caused disease. The scientific importance of the manuscript is very low as more specialized papers and review has been published on the subject, particularly for Xylella diseases. However, the major problem of this manuscript is the English. The language is poor, with a lot of grammatical mistakes as third persons of verbsand among with between, among the others. the sentences are very long, including different concept. Structure and word of the sentences as simple translation from the Italian languages. The top was reached with the use of "agrums" or "lymph". I therefore am not in favor of its publication. I am attaching a pdf file with all the corrections in the revision mode.

Reviewer 2 Report
This interesting review is devoted to the role of anatomy of vessels of woody plants in X. fastidiosa resistance. The paper is scientifically significant and timely; the list of literature cited is comprehensive and contains all the references coppesponding to the subject.
The only thing I would like to recommend to the authors is to read the MS text once again or use any English editing service to improve the style of some sentences.
Reviewer 3 Report
The manuscript entitled “Xylem embolism and pathogens: can vessels anatomy of woody plants contribute to X. fastidiosa resistance?” reviews literature regarding xylem features (physical and mechanical) that can determine plant tolerance to pathogens.
Review contains the required and updated information with large number of required references related to the topic. Overall, I find the review very comprehensive.
I only have minor comments/suggestions for the authors:
Line 48: please remove (Wells ) from the main text, as it is already cited with [14]
Line 123: I suggest to remove the comparison “than plants with long…”. It is completely understood regarding the context of the statement.
Line 187 and 209: a letter “s” is missing in “act” and “belong”
Line 191: “Infact” should be change to “In fact”
Line 303: Remove one dot.
Line 305: Rephrase “can be due to because can maintenance…”. One suggestion might be: “can be due to the maintenance of the...”
Table 1: I would suggest to place the table in one page to facilitate the lecture.
In lines 341-344 you suggested that olive tree can manage drought by reducing vessels diameter and increasing vessels density, and also in lines 362-364 you referenced as Leccino with higher number of vessels with smaller diameter suggesting its resistance to Xylella fastidiosa. Therefore, how can you differentiate the vessels anatomy under the presence of abiotic (drought) or biotic (pathogen) stresses? Also, you mentioned that Cellina di Nardò presents higher susceptibility to Xylella fastidiosa due to the larger diameter of xylem vessels that facilitate its spread. So, would be Cellina di Nardò more tolerant to the pathogen under drought conditions that reduce the xylem vessels diameter?
If xylem vessels anatomy could be a good strategy to contribute to the pathogen resistance in woody plants as it is indicated in the tittle of the manuscript, how can we manage to modify them in order to reach that resistance? Would it be that possible?
Line 513: doi is missing in references 29, 31, 32, 36, 37, 44, 50, 175, 178, 190 and 191. Also include the full reference in 191.
Round 2
Reviewer 1 Report
None, I agree to publish the review.
Reviewer 2 Report
The revised version of the paper is scientifically sound and interesting for the specialists. I believe the MS can be accepted in the current form.
Reviewer 3 Report
I consider the authors have addressed all my concerns and I believe this manuscript is ready for publication.
